# Production of Reactive Oxygen (ROS) and Nitrogen (RNS) Species in Macrophages J774A.1 Activated by the Interaction between Two *Escherichia coli* Pathotypes and Two Probiotic Commercial Strains

**DOI:** 10.3390/microorganisms11071644

**Published:** 2023-06-23

**Authors:** Berenice González-Magallanes, Fátima Sofía Magaña-Guerrero, Victor Manuel Bautista-de Lucio, Jimmy Giovanni Hernández-Gómez, Angel Gustavo Salas-Lais, Humberto Hernández-Sánchez

**Affiliations:** 1Unidad de Investigación del Instituto de Oftalmología “Fundación de Asistencia Privada Conde de Valenciana I.A.P.”, Chimalpopoca 14, Obrera, Mexico City 06800, Mexico; bgonzalezm1003@alumno.ipn.mx (B.G.-M.); fatima.magana@institutodeoftalmologia.org (F.S.M.-G.); vbautistal@institutodeoftalmologia.org (V.M.B.-d.L.); 2Departamento de Ingeniería Bioquímica, Escuela Nacional de Ciencias Biológicas, Instituto Politécnico Nacional, Unidad Profesional Adolfo López Mateos, Wilfrido Massieu Av. C.P., Mexico City 07738, Mexico; 3Coordinación de Calidad de Insumos y Laboratorios Especializados, Instituto Mexicano del Seguro Social, Mexico City 07760, Mexico

**Keywords:** probiotics, paraprobiotics, immunobiotics, postbiotics, macrophages, *E. coli* Nissle 1917, *Bifidobacterium*, ROS, RNS

## Abstract

Probiotics play an important role against infectious pathogens, such as *Escherichia coli* (*E. coli*), mainly through the production of antimicrobial compounds and their immunomodulatory effect. This protection can be detected both on the live probiotic microorganisms and in their inactive forms (paraprobiotics). Probiotics may affect different cells involved in immunity, such as macrophages. Macrophages are activated through contact with microorganisms or their products (lipopolysaccharides, endotoxins or cell walls). The aim of this work was the evaluation of the effect of two probiotic bacteria (*Escherichia coli* Nissle 1917 and *Bifidobacterium animalis* subsp. *lactis* HN019 on macrophage cell line J774A.1 when challenged with two pathogenic strains of *E. coli*. Macrophage activation was revealed through the detection of reactive oxygen (ROS) and nitrogen (RNS) species by flow cytometry. The effect varied depending on the kind of probiotic preparation (immunobiotic, paraprobiotic or postbiotic) and on the strain of *E. coli* (enterohemorrhagic or enteropathogenic). A clear immunomodulatory effect was observed in all cases. A higher production of ROS compared with RNS was also observed.

## 1. Introduction

*Escherichia coli* (*E. coli*) is one of the first bacteria to colonize the human gut (just a few hours after birth), forming part of the host microbiota and generally obtaining benefits from their relationship. These commensal strains seldom cause disease unless the host is immunocompromised or its gastrointestinal barrier integrity is lost [1]. However, there are also pathogenic variants which can be classified into diarrheagenic and extraintestinal infective *E. coli* which include diverse pathotypes and several natural hybrid strains. These variants can be obligate or facultative pathogens. The obligate pathogenic variants are able to cause disease in different conditions from moderate diarrheic episodes to fatal outcomes. The facultative variants, on the other hand, can behave as commensal bacteria in the intestinal tract but act as opportunistic pathogens when outside of this environment. The diarrheagenic *E. coli* can be classified into nine human pathotypes based on virulence factors [2]: Shiga toxin-producing (STEC), enterohemorrhagic (EHEC), enteropathogenic (EPEC), enteroaggregative (EAEC), enteroinvasive ((EIEC), enterotoxigenic (ETEC), diffusely adhering (DAEC), adherent-invasive (AIEC), and cell-detaching (CDEC) *E. coli*.

The research on probiotics began over one hundred years ago when they were first known to have potential health benefits. The International Scientific Association for Probiotics and Prebiotics (ISAPP) defined probiotics as “live microorganisms that, when administered in adequate amounts, confer a health benefit on the host” [3]. Among the many potential health benefits, the antimicrobial effect against many pathogenic microorganisms is well known. There are reports of this effect against *E. coli*, *Bacillus cereus* [4], *Vibrio cholerae* [5], *Salmonella enterica* Serovar Typhimurium [6], *Helicobacter pylori* [7], *Listeria monocytogenes* [8], *Pseudomonas aeruginosa* [9], *Staphylococcus aureus* [10], *Candida glabrata* [11], etc. There are many variations among the inhibition mechanisms including the release of organic acids, enzymes, small peptides with antimicrobial and antiviral activity, and bacteriocins by the probiotics. Bacteriocins are able to modulate the autochthonous microbiota and immunity of the host [12,13]. The secondary metabolites derived from the probiotic metabolism (vitamins, amino acids, or antimicrobial compounds), known as postbiotics, have recently gained attention due to their potential beneficial effects in humans such as in the prevention or treatment of conditions such as metabolic syndrome, neurological disorders or cancer [14]. It is well documented that probiotics exert immunomodulatory effects mainly at the level of the gut mucosa [15]; however, many reports have indicated that sometimes many different types of microbe-derived substances such as inactivated cells or their extracts and cell wall fragments, known as paraprobiotics, may also have beneficial physiological effects [16,17,18,19] at different levels, including on the immune system.

Specific strains of probiotics can act in the gut environment, mainly in the mucosa and intestinal barrier, to exert their immunomodulatory effect. Probiotics may affect many cells involved in immunity such as dendritic, epithelial and natural killer cells, monocytes, macrophages and lymphocytes, where activation occurs via pattern recognition receptors (PRRs) expressed in immune (M cells and Peyer’s patches) and non-immune (intestinal epithelium) cells. Toll-like receptors are well known PRRs able to activate signaling pathways responsible for cellular proliferation and the release of key cytokines, in order to, directly and indirectly, regulate cell-mediated immunity [20]. Macrophages are activated by the direct contact with microorganisms and/or their products (lipopolysaccharides, endotoxins or cell walls) [21]. Reactive Oxygen Species (ROS) and Reactive Nitrogen Species (RNS) achieve crucial functions in immunity, such as the initiation of cytocidal reactions involved in the pathogen defense tactic, and the induction and outlining of the immune response. During the development of infection conditions, pathogens are potent inducers of RNS-ROS formation in target cells like macrophages. Immune-regulatory pathways are started during T helper cell type 1 immune response associated with RNS-ROS production by immunocompetent cells. Therefore, the detection of these reactive species can be used for the in vitro and in vivo monitoring of the activation of the cellular immune system [22]. To the best of our knowledge, there are very few works about the activation of macrophages by probiotics or their preparations.

In our work, two commercial probiotics commonly used as treatment for intestinal infections (including *Escherichia coli* pathotypes) were used, with the main characteristic that they are antigenically distant: on the one hand, the Gram-positive bacterium *Bifidobacterium animalis* subsp. *lactis* HN019 and, on the other hand, the Gram-negative bacterium *Escherichia coli* Nissle 1917. Despite the multiple evidence of the health benefits of these probiotics, little is known about how they regulate inflammatory processes such as oxidative stress.

The aim of this work was then to evaluate the effect of two commercial probiotic bacteria (*Escherichia coli* Nissle 1917 and *Bifidobacterium animalis* subsp. *lactis* HN019 on the production of RNS and ROS by the macrophage cell line J774A.1 when challenged with two pathogenic strains of *E. coli*.

## 2. Materials and Methods

### 2.1. Macrophage Cell Line and Growth Conditions

J774A.1 is a cell line isolated from the ascites of an adult, female mouse BALB/c with reticulum cell sarcoma. Cells were cultured in T25 bottles at 37 °C in a humidified chamber containing 5% CO_2_ in F12K medium (Gibco) supplemented with fetal bovine serum (FBS, Gibco) and 0.01% Gibco™ Penicillin-Streptomycin (10,000 U/mL). Cells were subcultured until a minimal confluence of 90% was achieved as determined by light microscopy. After that, the adherent cells were mechanically removed and their number determined microscopically with a Neubauer chamber and their viability evaluated by Trypan blue vital staining [23].

### 2.2. Bacterial Strains and Growth Conditions

Two commercial probiotic bacterial strains were used in this work: *Escherichia coli* Nissle 1917 (EcN) isolated from the commercial supplement Mutaflor^®^ (Pharma-Zentrale, Herdecke, Germany) and the commercial strain Howaru^®^ *Bifidobacterium animalis* subsp. *lactis* HN019 (IFF, New York, NY, USA) (HN019). EcN and HN019 were cultured (37 °C) and maintained in Luria-Bertani (LB) and de Man–Rogosa–Sharpe (MRS) modified with 0.05% cysteine media, respectively. The pathogenic *E. coli* strains included an EHEC O157:H7 (EDL933) and an EPEC (E2348/69) and were cultured at 37 °C and maintained in LB medium. The cells from 24 h cultures of all the strains were centrifuged and suspended in sterile PBS (pH 7.2) before being used in the experiments [24].

### 2.3. Obtention of the Probiotic Preparations for Stimulation of the Macrophages

The elaboration of the three types of stimulating preparations for macrophages from EcN and HN019 was as follows [18]:(a)Postbiotics (PB). The culture suspensions were filter-sterilized (0.2 µm), and the cell-free filtrate (postbiotic preparation) kept for the stimulation experiments.(b)Immunobiotics (IB). The word “immunobiotic” has been proposed to describe microbial strains able to beneficially regulate the immune system. The cells from the previous experiment were suspended in sterile PBS (pH 7.2) and kept for the stimulation experiments after a viable count was performed.(c)Paraprobiotics (PP). The cells from the previous experiment were inactivated by heating at 100 °C/30 min and freezing at −20 °C/5 min. This preparation was used for the stimulation experiments.

### 2.4. Macrophage Stimulation by Probiotic and Pathogenic Bacterial Strains

The macrophages (1 × 10^6^) were stimulated with IB or PP and EHEC or EPEC (1 × 10^7^). In the case of PB, 40 µL were used for the stimulation. The assays were performed directly in the cytometry tubes due to the nature of the macrophages [25].

Two types of bacterial challenge were applied to the macrophages:(a)The macrophages were stimulated with the three types of probiotic-derived preparations (IB, PP or PB) for 24 h at 37 °C in a 5% CO_2_ environment. After this, the cells of EHEC or EPEC were added to the system and incubated for another 6 h.(b)In this second case, the macrophages were stimulated with the cells of EHEC or EPEC for 6 h at 37 °C in a 5% CO_2_ environment. After this, the probiotic-derived preparations (IB, PP or PB) were added to the system and incubated for another 24 h at 37 °C. A negative control containing PBS only was also used in both challenges.

### 2.5. Measurement of the Production of Reactive Oxygen (ROS) and Nitrogen (RNS) Species

The kit CellROX^®^ Deep Red (Invitrogen, Waltham, MA, USA) for flow cytometry assay was used for the determination of ROS and the dye (4-amino-5-methylamino-2′,7′-difluorofluorescein) diacetate (DAF-FM DA) (Thermo Scientific, Eugene, OR, USA) was used to determine RNS. The fluorescent probes were used at a 5 µM concentration for labeling purposes. For this reason, the macrophages, bacteria and supernatants were suspended in a 98 µL final volume [26,27]. The labeling procedure was started 60 min before the end of the incubation time for each assay. First, the CellROX^®^ Deep Red (5 µM) probe was added and the tubes were incubated for 30 min at 37 °C in 5% CO_2_ in the dark. After that, the DAF-FM DA probe (5 µM) was added to the tubes and they were incubated for another 30 min. A separated tube of macrophages with the same assay conditions were labeled with 4 μL of conjugated-APC monoclonal antibody anti-mouse F4/80 (Thermo Scientific, Eugene, OR, USA), incubating this for 30 min in darkness [28]. The analysis was performed on a BD FACSLyric™ Flow Cytometry System (Figure 1).

### 2.6. Statistical Analysis

All the experiments were run in triplicate and the data are expressed as the mean ± standard error of the mean. Statistical difference between the means was determined by one-way analysis of variance (ANOVA) followed by a Tukey post hoc test at a significance level of α = 0.05. The analyses were performed using GraphPad Prism version 8.00 for Windows (GraphPad Software, San Diego, CA, USA).

## 3. Results

The measurement of both reactive species was carried out by flow cytometry. The results were grouped by reactive species and by probiotic strain. The medium fluorescence intensity (MFI) was calculated from the individual readings in the flow cytometer. The data for both bacterial challenges are included (see Section 2.4).

### 3.1. Determination of RNS

The results of the MFI for the RNS, where the probiotic EcN and the pathogenic EHEC and EPEC strains were used, are shown in Figure 2. A significantly higher production of RNS (*p* ≤ 0.05) can be observed (Figure 2a) in the case of the stimulation of the macrophages with EHEC. The other combinations resulted in very low values of RNS (similar to the value of the negative control). In the case of EPEC (Figure 2b), no significant differences (*p* > 0.05) could be detected between the bacterial challenges, although their values were significantly higher than the negative control. In the case of the EcN paraprobiotic experiments (Figure 2c), a significantly lower RNS production was detected when the macrophages were stimulated with the EHEC compared with the challenges with PP and PP-EHEC. A similar trend was observed in the case of the EcN PP and EPEC (Figure 2d). In the case of the EcN postbiotics experiments, the higher RNS production was obtained in the runs in which the initial stimulation was performed with the PB, followed by the pathogen (see Figure 2e,f). The results of the MFI for the RNS using HN019 as the probiotic and EHEC (Figure 3a) and EPEC (Figure 3b) as the pathogens were similar since the highest amount of RNS was produced when the immunobiotic preparation was used for the challenge. Figure 3c,d shows that the highest amount of RNS was produced when the paraprobiotic preparation was used for the challenge for both strains of *E. coli*. Finally, Figure 3e,f indicates that the highest amount of RNS was produced when the postbiotic preparation was used for the challenge before the stimulation of the macrophages with the pathogenic strains of *E. coli*.

### 3.2. Determination of ROS

The results of the MFI for the ROS, where the probiotic EcN and the pathogenic EHEC and EPEC strains were used, are shown in Figure 4. Figure 4a,b indicates that the highest amount of ROS was produced when the immunobiotic preparation was used for the challenge before the stimulation of the macrophages with the pathogenic strains of *E. coli*. In the case of the EcN paraprobiotic, the highest amount of ROS was produced when the pathogen was the only stimulus for the macrophages, followed by the process in which the paraprobiotic preparation was used for the challenge before the stimulation of the macrophages with the pathogens. In the experiments with the EcN postbiotic, the highest ROS production was obtained for the pathogen and postbiotic-pathogen challenges for EHEC (Figure 4e) and for the pathogen, postbiotic, and postbiotic-pathogen challenges for EPEC.

Figure 5 shows the results of the MFI for the ROS when the probiotic HN019 and the pathogens EHEC and EPEC were used for macrophage stimulation. Figure 5a shows a highest production of ROS in the cases where the HN019 immunobiotic and EHEC were used as single stimulus for the macrophages. In the case of EPEC, the four challenges induced a similar ROS production when used to stimulate the macrophage cell line (see Figure 5b). When the HN019 paraprobiotic was used, the results (Figure 5c,d) were similar to those observed in the case of the HN019 immunobiotic. Similar increases were observed when HN019 postbiotic and the EHEC and EPEC strains were used as stimuli for the macrophages. No significant differences were detected among the four kinds of challenges for both *E. coli* strains (see Figure 5d,e).

## 4. Discussion

Reactive oxygen species (ROS) are essential for the elimination of invasive microorganisms by macrophages. NADPH oxidase is the first source of ROS that has been recognized in macrophages. However, there is evidence that indicates that mitochondria are a critical site of ROS formation in macrophages. ROS comprises superoxide anion (O_2_^−^), hydrogen peroxide (H_2_O_2_), hydroxyl radical (OH), and singlet oxygen. Besides ROS, there are other reactive signaling molecules such as reactive nitrogen species (RNS), which include nitric oxide (NO) [27]. This last compound has many roles in immunity which include toxicity towards pathogens, apoptosis modulation in leucocytes, and immunoregulation. NO is often produced in response to a bacterial infection [28]. Considering this, the effect of different bacterial-derived stimuli on the production of RNS and ROS by macrophages J774A.1 was investigated. These stimuli included direct contact with pathogenic strains of *E. coli* (EHEC and EPEC) and probiotic strains (EcN and HN019) including their derivatives (immunobiotics, paraprobiotics and postbiotics). Two other kinds of challenges were also included: (1) The macrophages are first stimulated with a pathogenic strain and then with a probiotic preparation, and (2) the macrophages are first stimulated with a probiotic preparation and then with a pathogen. The generation of RNS and ROS was detected with specific flow cytometry techniques. A great variability in the results could be observed since the production of reactive species depends on the pathogenic strain, the probiotic strain and its derivatives and also on the kind of challenge used in the experiment.

In this study, activation of the macrophages by some of the immunobiotics could be observed. The production of ROS and RNS was higher with the HN019 immunobiotic for both pathogenic strains. It is well known that the induction of the enzyme nitric oxide synthase (NOS) can be achieved by means of different signals such as lipopolysaccharides (LPS), mannose, CpG dinucleotides, and lipoteichoic acid present in several bacterial groups. This last compound is present in the cell wall of *Bifidobacterium animalis* subsp. *lactis* and is considered a postbiotic [29]. The components of the bacterial surface of bifidobacterial have an important role as immunomodulating agents [30]. This is in agreement with the results obtained for HN019 (Figure 3a,b). However, in the case of EcN, this was not true, since higher values of RNS were obtained when the pathogenic strains were used by themselves or in the combined challenges. It is well known that EcN has a serotype O6 LPS, which is very different from other LPS by having a very short side chain formed by a single repetitive unit in the oligosaccharide part of the molecule [31]. These variations make the EcN very immunogenic but not immunotoxic. This fact might explain why the generated level of RNS in the challenges IB-EHEC and EPEC-IB are lower compared with the levels generated when EHEC or EPEC were used as the only stimulus. It is very likely that, in this case, the EcN immunobiotic works as an immunoregulator by lowering the RNS levels. Salas-Lais et al. [28] demonstrated that an immunobiotic preparation of *Lacticaseibacillus casei* Shirota was able to activate macrophages and produce NO. A similar behavior was observed in the case of the production of ROS in the IB-EHEC and IB-EPEC challenges and again, a higher production was obtained when EHEC or EPEC was used as the only stimulus. However, in the case of the challenges EHEC-IB and EPEC-IB, the levels of ROS were the highest, indicating that, in this case, when the macrophages are preactivated with the pathogenic strains, the final response is higher. A suitable explanation must be found for this phenomenon. In the instance of HN019, the four challenges induced lower levels of ROS than when EHEC or EPEC was used as the only stimulus, indicating that the HN019 immunobiotic works also as an immunoregulator of the levels of ROS. The difference in behavior could lie in the synthesis and activation mechanisms. The NADPH oxidases (NOX1to 5 and DUOX1 to 2) are membrane-associated enzymes which use NADPH as an electron donor to induce the formation of the superoxide anion and H_2_O_2_ from molecular oxygen. NOX2 is the main source of ROS production [32] and its activation and assembly start with a signalization event after the stimulation of receptors for formylated peptides, C5a, Fc or pattern-recognition receptors such as TLR4 [33]. In these systems, NO is synthesized by an inducible NOS after activation with LPS, cytokines, or other molecules. Once activated, the NOS has a constant activity which is independent of the intracellular concentration of Ca [34]. The synthesis of RNS is definitely more complex than that of the ROS.

As mentioned before, paraprobiotics play an important role in the immune system activity since dead cells have anti-inflammatory effects in the gastrointestinal tract [35]. The consumption of probiotics and paraprobiotics is beneficial since they are able to modulate the microbiota and induce the expression of genes involved in the intestinal immune response [36]. The components in the paraprobiotics vary with the strain of the probiotic, but in general include peptidoglycans, surface proteins, cell wall polysaccharides, etc. [18]. The purpose of using paraprobiotics in this work was to compare the effect of this components in macrophage activation in live and dead cells. In this study, the EcN paraprobiotics were able to activate the macrophages producing higher levels of RNS than ROS, whereas the HN019 paraprobiotics stimulate the production of similar levels of RNS and ROS. HN019 paraprobiotics were more effective in the stimulation of reactive species production than the immunobiotics. This could be due to the generation of new structures (denaturation of proteins and DNA) during the heat treatment process [37]. In other studies, it was shown that the *L. casei* Shirota paraprobiotics are able to activate macrophages so they can produce nitric oxide [28] and that *L. casei* IMAU60214, *Lacticaseibacillus rhamnosus* GG and KLDS and *Lactobacillus helveticus* IMAU70129 are able to activate macrophages to produce ROS [25]. These studies corroborate the fact that many probiotics and paraprobiotics are able to induce the production of RNS and ROS. However, there are also other mechanisms to generate reactive species. The gut microbiota can also directly generate ROS and RNS. Macrophages can induce *Enterococcus faecalis* to create superoxide hydroxyl radicals [38].

In the case of the postbiotics, in all cases the macrophage activation induced a similar production of reactive species compared with all the other experiments. The amount of RNS was higher than ROS. It is possible that mechanisms of immune regulation are involved when postbiotics are in the system. There are reports that bifidobacteria (including HN019) are able to produce lactate and acetate as the final products of fermentation [39] and that acetate can stimulate the macrophages to improve their bactericidal activity. Acetate initiates a transcriptomic program in macrophages that induces changes in the metabolic process and in the exits of the immunity effectors [40]. There are relatively few papers dealing with macrophage activation by *E. coli*. [23,41,42] and in all cases only the positive ROS generation was measured.

Macrophage activation was carried out using different stimuli (probiotic preparations, pathogens, and combination challenges) which in turn resulted in different amounts of RNS and ROS as was shown in this work. It is known that when macrophages are activated by interferons (IFN) or LPS, the phenotype M1 results. On the other hand, when the activation is performed by interleukins (IL-4 or IL-13), immune complexes, or glucocorticoids, the phenotype M2 is the result [43,44]. The M1 macrophages have an inflammatory role and are generally activated by intracellular pathogens, LPS, IFN-γ and TNF-α among others. The activation process includes high levels of proinflammatory cytokines, mechanisms of pathogen resistance, a strong microbicidal and tumoricidal activity, a high production of RNS and ROS and the promotion of Th1 response. M2 macrophages are anti-inflammatory. There are reports indicating that the LPS from the outer membrane of Gram-negative bacteria plays an important role in chronic infections but if the infection lingers too long, M1 macrophages are polarized to the M2 phenotype to keep a tolerance state to LPS [45].

Different probiotic strains are able to stimulate the macrophages in different ways to an M1 or M2 polarization which involves many pathways and transcription factors. However, there are many reports that indicate that probiotics usually induce M1 macrophages to have a proinflammatory activity, but the molecular mechanisms involved in the regulation of the polarization process are still unclear. [46]. It is also known that when the stress is very high, some pathogenic strains use the ROS for self-destruction. Up to now, no protein-based mechanism has been discovered to eliminate the hydroxyl radical [47].

Postbiotics and paraprobiotics, as they are inert probiotic derivatives and do not have the ability to transfer or acquire genes, are a good option for the treatment of diseases of a different nature, since the beneficial protective effect that they can have (such as for example processes of inflammatory regulation and oxidative stress). In the health sector, it can play a very important role, incorporating it into the treatment of patients with pathologies with inflammatory processes (infectious and non-infectious)”.

## 5. Conclusions

The results indicate that the activation of the J774A.1 macrophages measured through the generation of RNS and ROS depends on the nature of the stimuli (probiotic strain and its immunobiotic, paraprobiotic and postbiotic preparation and the pathogenic strain used) and the type of challenge used in the activation protocol. Immunoregulatory activity was detected in the immunobiotics and paraprobiotics of both probiotic strains. In most of the experiments, the amount of RNS was higher than that of ROS, indicating the presence of different activation mechanisms and a certain selectivity.

## Figures and Tables

**Figure 1 microorganisms-11-01644-f001:**
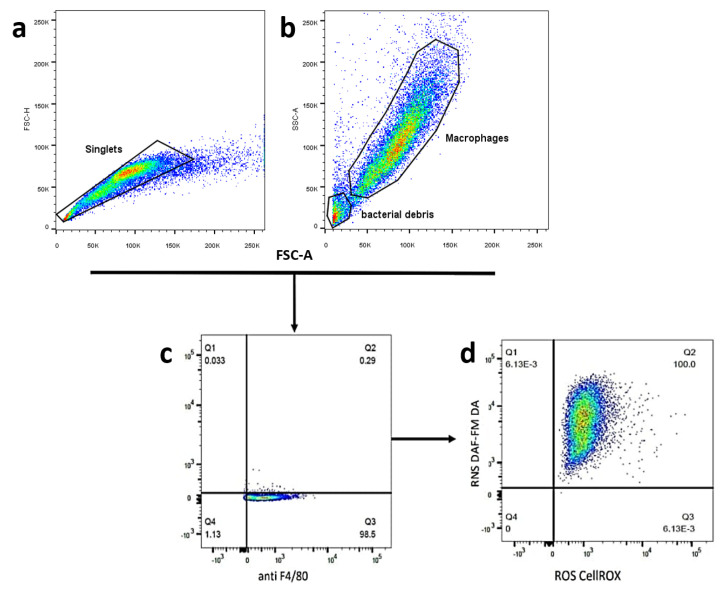
Flow cytometry gating strategy in macrophages J774A.1. (**a**) Singlets using FSC-H versus FSC-A. (**b**) Macrophages and bacterial debris separation using FSC-A versus SSH-A for the discrimination of cells by size and complexity. (**c**) Macrophage detection using the specific anti-mouse F4/80 antibody. (**d**) Finally, reactive oxygen species (ROS) and reactive nitrogen species (RNS) detection by double-labeling with CellROX and DAF-M DA, respectively, in the same sample.

**Figure 2 microorganisms-11-01644-f002:**
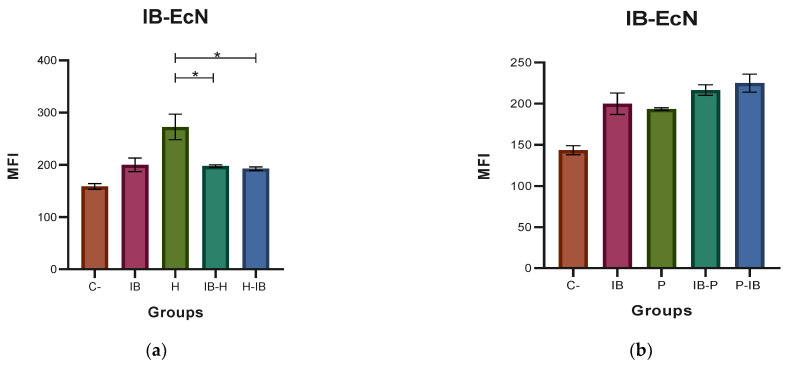
Medium fluorescence intensity (MFI) of reactive nitrogen species (RNS) with *Escherichia coli* Nissle 1917 (EcN). (**a**,**b**) MFI values of the two types of stimulation with immunobiotic (IB); (**c**,**d**) MFI values of the two types of stimulation with paraprobiotic (PP); (**e**,**f**) MFI values of the two types of stimulation with postbiotic (PB) of EcN against *Escherichia coli* EHEC (H) and EPEC (P) on J774A.1 macrophages were obtained. Results are expressed as the Mean ± SEM. * *p* < 0.05; ** *p* < 0.01; *** *p* < 0.001; **** *p* < 0.0001.

**Figure 3 microorganisms-11-01644-f003:**
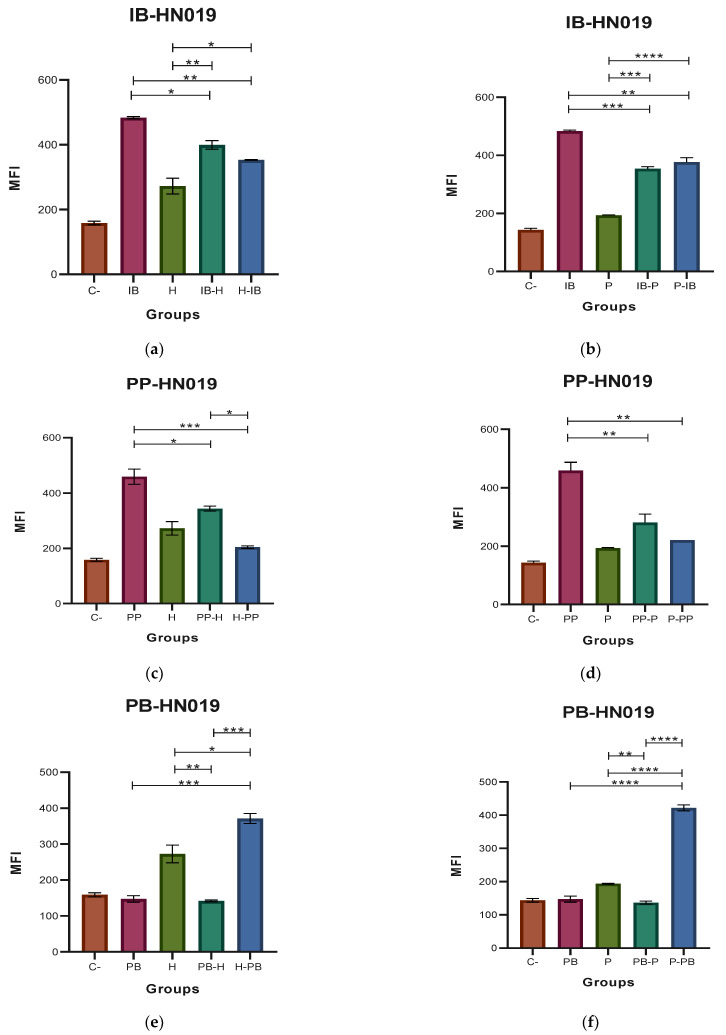
Medium fluorescence intensity (MFI) of reactive nitrogen species (RNS) with *Bifidobacterium animalis* subsp. *lactis* HN019 (HN019). (**a**,**b**) MFI values of the two types of stimulation with immunobiotic (IB); (**c**,**d**) MFI values of the two types of stimulation with paraprobiotic (PP); (**e**,**f**) MFI values of the two types of stimulation with and postbiotic (PB) of HN019 against *Escherichia coli* EHEC (H) and EPEC (P) on J774A.1 macrophages were obtained. Results are expressed as the Mean ± SEM. * *p* < 0.05; ** *p* < 0.01; *** *p* < 0.001; **** *p* < 0.0001.

**Figure 4 microorganisms-11-01644-f004:**
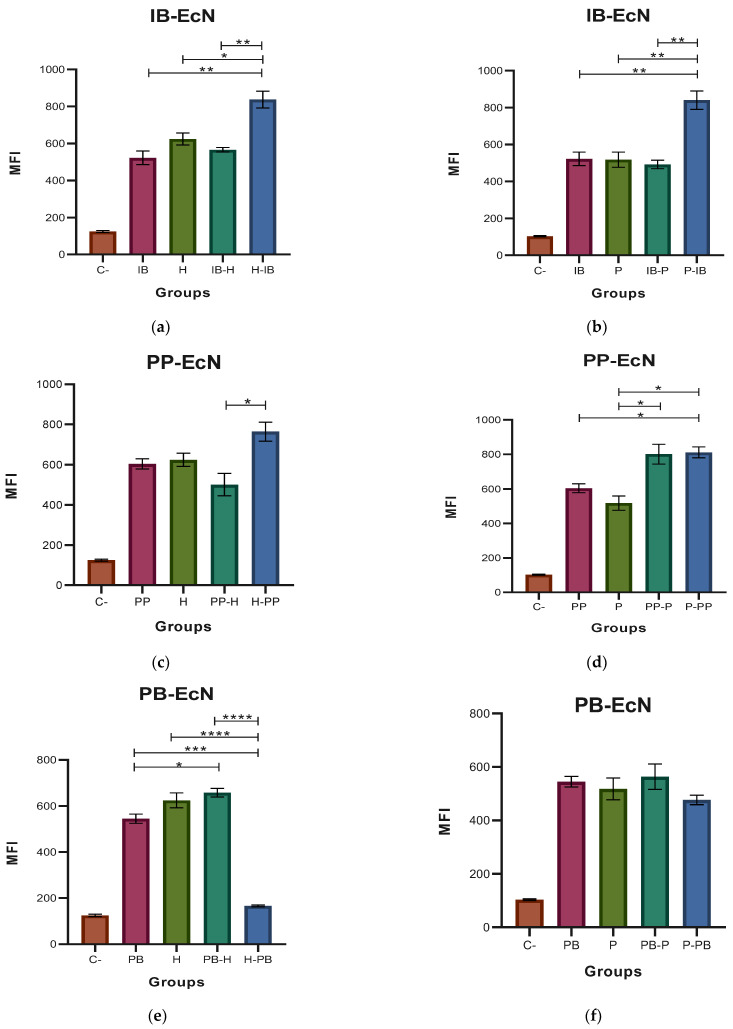
Medium fluorescence intensity (MFI) of reactive oxygen species (ROS) with *Escherichia coli* Nissle 1917 (EcN). (**a**,**b**) MFI values of the two types of stimulation with immunobiotic (IB); (**c**,**d**) MFI values of the two types of stimulation with paraprobiotic (PP); (**e**,**f**) MFI values of the two types of stimulation with and postbiotic (PB) of EcN against *Escherichia coli* EHEC (H) and EPEC (P) on J774A.1 macrophages were obtained. Results are expressed as the Mean ± SEM. * *p* < 0.05; ** *p* < 0.01; *** *p* < 0.001; **** *p* < 0.0001.

**Figure 5 microorganisms-11-01644-f005:**
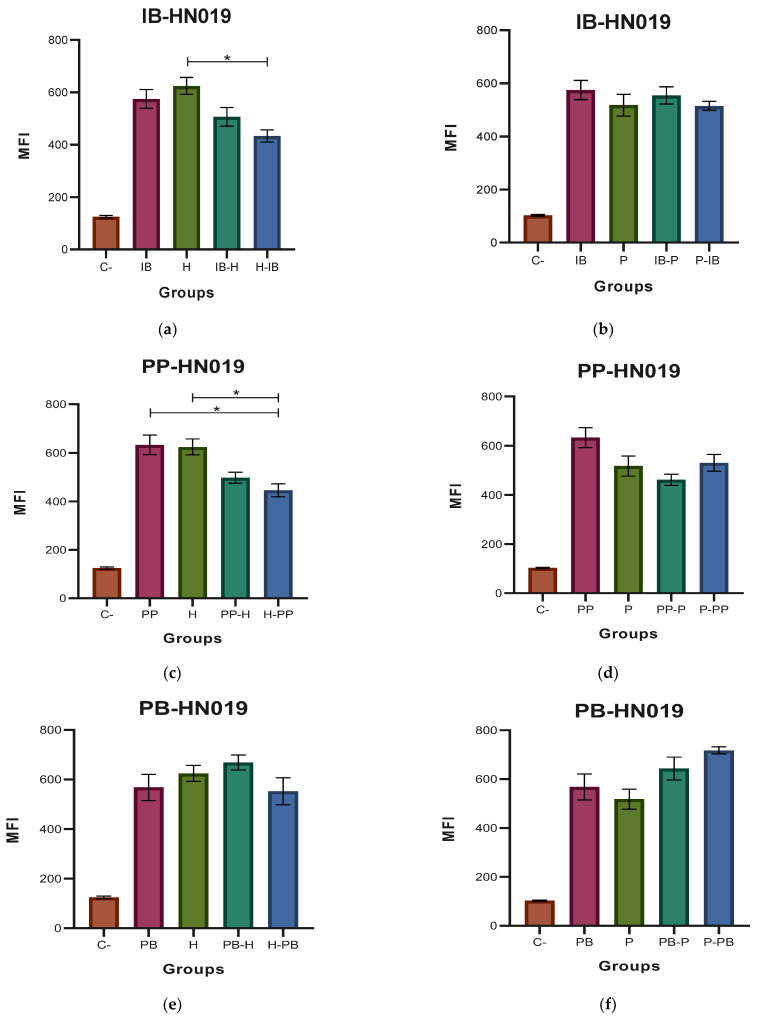
Medium fluorescence intensity (MFI) of reactive oxygen species (ROS) with *Bifidobacterium animalis* subsp. *lactis* HN019 (HN019). (**a**,**b**) MFI values of the two types of stimulation with immunobiotic (IB); (**c**,**d**) MFI values of the two types of stimulation with paraprobiotic (PP); (**e**,**f**) MFI values of the two types of stimulation with postbiotic (PB) of HN019 against *Escherichia coli* EHEC (H) and EPEC (P) on J774A.1 macrophages were obtained. Results are expressed as the Mean ± SEM. * *p* < 0.05.

## Data Availability

Data will be available upon justified request.

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
