# Peer review of "Production of Reactive Oxygen (ROS) and Nitrogen (RNS) Species in Macrophages J774A.1 Activated by the Interaction between Two Escherichia coli Pathotypes and Two Probiotic Commercial Strains"

_microorganisms, 2023, doi:10.3390/microorganisms11071644_

Round 1

Reviewer 1 Report

The paper suggested the activation of the J774A.1 macrophages measured through the generation of RNS and ROS depends on the nature of the stimuli (probiotic strain and its immunobiotic, paraprobiotic and postbiotic preparation and the pathogenic strain used) and the type of challenge used in the activation protocol. The research showed significance in the development of immune agents. Mostly, it can be accepted. However, there are still few questions need author's explanation.

1. As known, lactobacillus is a kind of probiotic and studied widespread. The author used Escherichia coli Nissle 1917 and Bifidobacterium animalis subsp. lactis HN019 in this study. I think the reason should be added in the part of "Introduction". 

2. Page 3, line 114, Part 2.3, there is a doubt about the definition of "Postbiotics (PB)" . Please make sure it is "cell supernatant" or "PBS cell suspensions".

3. Page 3, line 125, Part 2.4, why did the author choose 40μL PB to stimulate macrophages. Is there some data support? or other reasons?

4. The author described the immunoregulatory activities of probiotic strain and its immunobiotic, paraprobiotic and postbiotic preparation and the pathogenic. But the research significance is not introduced clearly.    

Minor editing of English language required.

Author Response

Thank you for reviewing our manuscript for possible publication on Microorganisms. We are very pleased to have the opportunity to revise our manuscript, and appreciate your insights.

  1. As known, lactobacillus is a kind of probiotic and studied widespread. The author used Escherichia coli Nissle 1917 and Bifidobacterium animalis lactis HN019 in this study. I think the reason should be added in the part of "Introduction". 

R= We appreciate the observation made and the information requested in the Introduction section (lines 87-93) was added: "In our work, two commercial probiotics commonly used as treatment for intestinal infections (including Escherichia coli pathotypes) were used, with the main characteristic that they are antigenically distant, on the one hand the Gram positive bacterium Bifidobacterium animalis subsp. lactis HN019 and, on the other hand, the Gram-negative bacterium Escherichia coli Nissle 1917. Despite the multiple evidence of the health benefits of these probiotics, little is known about how they regulate inflammatory processes such as oxidative stress”.

  1. Page 3, line 114, Part 2.3, there is a doubt about the definition of "Postbiotics (PB)". Please make sure it is "cell supernatant" or "PBS cell suspensions".

R= We appreciate the observation made and the text is amended as follows: “Postbiotics (PB). The culture suspensions were filter-sterilized (0.2 µm), and the cell-free filtrate (postbiotic preparation) kept for the stimulation experiments.”

  1. Page 3, line 125, Part 2.4, why did the author choose 40 μL PB to stimulate macrophages. Is there some data support? or other reasons?

R= We appreciate the observation made. Until now, there are few studies related to the use of postbiotics in in vitro cell culture models, some authors have used 20, 50 and 100 μL, however, there is no established volume pattern. In our work we decided to use 40 μL of the postbiotic because it is in the range of volumes used in the literature and technically, it is a volume in which the probes used are adjusted to a concentration of 5 μM.

  1. The author described the immunoregulatory activities of probiotic strain and its immunobiotic, paraprobiotic and postbiotic preparation and the pathogenic. But the research significance is not introduced clearly.    
R. Nowadays, there is a large amount of information about probiotics and how they can provide a beneficial effect on human health, among which is immunomodulation and regulation of inflammatory processes such as oxidative stress, these processes are manifested in infectious and non-infectious pathologies, so the evidence demonstrated in the present work proposes the use of non-viable bacterial structures and metabolites of commercial probiotics that can be complementary in pharmacological therapies, with special use in immunologically vulnerable people. This paragraph could be inserted between lines 93 and 94 in the manuscript.

Betesho Babrud, R.; Kasra Kermanshahi, R.; Motamedi Sede, F.; and Moosavinejad, S. Z. The effect of Lactobacillus reuteri cell free supernatant on growth and biofilm formation of Paenibacillus larvae. Iran. J. Vet. Res. 2019, 20(3), 192–198.

Koohestani, M.; Moradi, M.; Tajik, H.; Badali, A. Effects of cell-free supernatant of Lactobacillus acidophilus LA5 and Lactobacillus casei 431 against planktonic form and biofilm of Staphylococcus aureus. Vet. Res. Forum 2018, 9(4), 301-306.  

Scillato, M.; Spitale, A.; Mongelli, G.; Privitera, G.F.; Mangano, K.; Cianci, A.; Stefani, S.; Santagati, M. Antimicrobial properties of Lactobacillus cell-free supernatants against multidrug-resistant urogenital pathogens. Microbiologyopen 2021, 10(2), e1173.

Reviewer 2 Report

This study on the impact of various probiotics, paraprobiotics, and postbiotics on macrophage activation and ROS and RNS production. This is important in the growing field of probiotic research and in understanding how the gut microbiota interacts with the immune system.

Minor Concerns:

The authors could have provided more explicit reasoning as to why the study was important, including potential applications of their findings in clinical or public health settings. While ROS and RNS production can provide some insights into macrophage activation and function, a more comprehensive and multifactorial approach is needed to accurately determine macrophage phenotypes.

Flow cytometry uses forward scatter to provide an estimate of cell size on a linear scale, and the reported sizes of E. coli debris and macrophages appear to be inconsistent with their known size differences. E. coli cells are significantly smaller than macrophages, typically around 0.5-2 micrometers in length, while macrophages are generally in the range of 10-30 micrometers in diameter. The flow cytometry data suggest that E. coli debris is up to or more than half the size of what was gated as macrophages, this would seem inconsistent with the known biology of these cell types.

Major Concern:

The simultaneous use of CellROX™ Deep Red Reagent and APC, both excited by the same laser line and having similar emission spectra, leads to significant spectral overlap. This makes it impossible to definitively differentiate the signals, potentially skewing conclusions about ROS levels and macrophage activation.

The paper is read well and there are minor mistakes that need to be taken care of. 

Author Response

Thank you for reviewing our manuscript for publication on Microorganisms. We are very pleased to have the opportunity to revise our manuscript, and appreciate your insights.

Minor Concerns:

The authors could have provided more explicit reasoning as to why the study was important, including potential applications of their findings in clinical or public health settings. While ROS and RNS production can provide some insights into macrophage activation and function, a more comprehensive and multifactorial approach is needed to accurately determine macrophage phenotypes.

R= We appreciate the observation made and add a text in the section 4. Discusión (lines 346-351): “Postbiotics and paraprobiotics, as they are inert probiotic derivatives and do not have the ability to transfer or acquire genes, are a good option for the treatment of diseases of a different nature, since the beneficial protective effect that they can have (such as for example processes of inflammatory regulation and oxidative stress). In the health sector, it can play a very important role, incorporating it into the treatment of patients with pathologies with inflammatory processes (infectious and non-infectious).”

As you properly told us, these results aren´t enough to provide information about macrophages phenotypes but in the future more studies could be made to know more about the macrophages polarization activity.

Flow cytometry uses forward scatter to provide an estimate of cell size on a linear scale, and the reported sizes of E. coli debris and macrophages appear to be inconsistent with their known size differences. E. coli cells are significantly smaller than macrophages, typically around 0.5-2 micrometers in length, while macrophages are generally in the range of 10-30 micrometers in diameter. The flow cytometry data suggest that E. coli debris is up to or more than half the size of what was gated as macrophages, this would seem inconsistent with the known biology of these cell types.

R= We thank you for this commentary, and we agree with the reviewer. The original Figure 1 shows a population of bacterial debris bigger that usually are. In the new version of Figure 1 these gates have been corrected, the first gating (FSH-H vs. FSH-A) shows the singlets events taken for the analysis, followed of the gate of complexity (SSC-A) vs. size (FSC-A) where the population of macrophages were selected, and the bacterial debris together with other cellular debris can be founded near the low left axis.

Major Concern:  

The simultaneous use of CellROX™ Deep Red Reagent and APC, both excited by the same laser line and having similar emission spectra, leads to significant spectral overlap. This makes it impossible to definitively differentiate the signals, potentially skewing conclusions about ROS levels and macrophage activation.  

R. Indeed, the signal of CellROX™ Deep Red Reagent and APC have a similar emission spectrum and this is why the F4/80 staining was not performed together with the ROS detection and the label against F4/80 was done in a separated tube of cells that were incubated under the same assay conditions.  In this sense to clarify, we rewritten this part in the methods section as follows: “A separated tube of macrophages with the same assay conditions were labeled with 4 μL of conjugated-APC monoclonal antibody anti‐mouse F4/80 (Thermo Scientific, Eugene, USA) incubating for 30 min in darkness”. See lines 157-160 of the new manuscript.

Round 2

Reviewer 2 Report

The author's responses to the comments were clear.